# Carotid Dolichoarteriopathy (Elongation) of the Carotid Arteries in Patients with Ischemic Stroke Anamnesis

**DOI:** 10.3390/biomedicines11102751

**Published:** 2023-10-11

**Authors:** Denis A. Golovin, Tatyana M. Rostovtseva, Yuri S. Kudryavtsev, Alexander B. Berdalin, Svetlana E. Lelyuk, Vladimir G. Lelyuk

**Affiliations:** 1Department of Clinical and Experimental Physiology of Circulatory System, Ultrasound and Functional Diagnostics of Federal State Budgetary Institution, Federal Center of Brain Research and Neurotechnologies of the Federal Medical Biological Agency Russian Federation, 123182 Moscow, Russia; rostovceva@fccps.ru (T.M.R.); kudriavcev.a@fccps.ru (Y.S.K.); vglelyuk@fccps.ru (V.G.L.); 2Federal State Budgetary Educational Institution, Further Professional Education “Russian Medical Academy of Continuous Professional Education”, Ministry of Healthcare of the Russian Federation, 127051 Moscow, Russia

**Keywords:** dolichoarteriopathy, carotid arteries elongation, tortuosity, pathological deformity, ischemic stroke, body mass index, arterial hypertension

## Abstract

Carotid artery elongation (ECA) is widespread in the asymptomatic population and among people with a history of ischemic stroke (IS). There are different points of view on the ways these changes contribute to brain ischemic damage pathogenesis. Materials and Methods: From 2019 to 2021, we included 1171 people who had suffered from IS less than one year before the investigation in the study, 404 (34.5%) women aged 27 to 95 years (64 ± 13 years) and 767 men (21–90; 60 ± 11 years). All patients involved in the study underwent multimodal radiological investigation in addition to assessments of their clinical and neurological data. Results: In this study, we were unable to detect a relationship between ECA localization and acute ischemic lesions. The frequency of ECA detection in patients with IS was the same as that in carotid and vertebral–basilar arterial systems. The prevalence of ECA was the same in patients with different IS subtypes (TOAST). There was no association between the localization of ECA and ischemic lesions; moreover, there were no differences in the IS frequency between anterior and posterior circulation. There were statistically significant decreases in linear peak systolic and end diastolic velocities in the internal carotid and vertebral arteries, as well as in the intracranial arteries in patients with ECA.

## 1. Introduction

The elongation of extracranial portions of internal carotid arteries (ECA) is widespread among the population. Its frequency in prospective population studies reaches 30%, while in hospital statistics it can be up to 58%, depending on the set of instrumental diagnostic methods [1]. There is no specific definition for this condition. It represents a significant deviation of the artery path from the linear. Due to the lack of clear diagnostic criteria, the use of various imaging modalities and the different rates of occurrence in people of different ages, the prevalence of dolichoarteriopathy can vary depending on the combination of these components. Most researchers suppose that ECA is based on an increase in artery length; therefore, the term “elongation” is used in this study. ECA may be congenital, due to the presence of genetically determined connective tissue disorders, as well as variants of dorsal and ventral aorta branch connections during ontogenesis [2,3,4,5]. Acquired or secondary-etiology ECA also exists, but has not been sufficiently studied. This condition can be associated with both an absolute increase in the ICA length, for example, with AH [6], and a relative one, associated with diseases or conditions accompanied by a change in the length of large vessels and vertebral structure and other bone ratios, usually arising from spinal pathology, obesity, etc. [7]. 

Some studies have been based on numerous patients of different ages who were investigated via ICA ultrasound duplex scanning [8]. The maximum ECA severity was shown (according to Metz’s classification [9]) in children and adolescents (in the age group of up to 20 years). In persons aged from 20 to 50 years, the frequency of registered ECA is small, but elongation frequency increases again for older patients [8]. Although ECA is common in children (in addition to dystrophic changes in the vertebral column not being usual in this cohort), spontaneous ECA “straightening” is frequently detected during repeated investigations.

Problems surrounding ECA research interest mainly exist due to some scientists’ opinions, according to which elongations are considered to be among the possible causes of IS [10,11,12]. In the presence of some conditions reflecting the severity of local blood flow disturbances in deformed areas of arteries, they can be indications of the necessity of surgical correction for the purpose of primary or secondary IS prevention both in adults and children [13]. This statement is posited in Part 3, “Brachiocephalic arteries”, of the National Guidelines for Management of Patients with vascular Arterial Pathology (Russian Consensus Document, 2013). At the same time, in contemporary Russian [14,15] and international recommendations on primary and secondary stroke prevention, there are no ECA mentions among the possible causes of IS and conditions requiring preventive correction, or information about the presence of reasonable indications for their surgical treatment [16,17].

This inconsistency is extremely important from both theoretical and practical points of view, since the focus is on the justification of preventive invasive surgical interventions. Thus, studies aimed at finding and verifying factors associated with ECA that could be considered evidence of their hemodynamic and/or pathogenetic significance are reasonable and relevant.

Many studies carried out by various teams have indicated some significant points that require further clarification, such as the presence, nature, severity, and causes of possible perfusion disorders [18], associations with ECA embolism in the distal circulation [19], and concomitant ECA conditions (dissections, etc.) [20,21].

However, to date, there is no generally accepted opinion as to whether or not ECA can be a cause of IS. The problem is aggravated by the pronounced etiopathogenetic heterogeneity of IS, and often the presence of several concurrent stroke causes. Our study aimed to investigate the presence, frequency, and localization of ICA elongation, and the hemodynamics of involved arteries in distal portions in ischemic stroke (IS) survivors.

In addition, we assessed the possible role of ECA in acute focal cerebral ischemia development, and tested the hypotheses of various factors leading to ECA formation.

## 2. Materials and Methods

The data of 1171 people who had suffered from IS no later than one year before inclusion, during the period from September 2019 to April 2021, were analyzed. All patients included in the study were hospitalized at the Federal State Budgetary Institution, “Federal center of brain research and neurotechnologies”, of the Federal Medical Biological Agency for rehabilitation by highly qualified specialists. The severity of neurological deficit averaged at 3 points, according to the mRS. The sample included 404 (34.5%) women aged from 27 to 95 years (average: 64 ± 13 years) and 767 men aged from 21 to 90 years (average: 60 ± 11 years). Of these, 951 patients exhibited IS in anterior circulation, and 220 people exhibited IS in posterior circulation (the ratio of IS frequencies in anterior circulation to posterior circulation was 4.3:1, which does not differ from the standard).

In all cases, anamnesis data, clinical neurological investigation results, and anthropometry with BMI calculations were collected [22]. In addition to assays of blood pressure level, heart rate, and multimodal instrumental examination, including assessments of vascular status and hemodynamics via duplex scanning (DS), computed tomographic angiography (CTA) and CT perfusion, structural changes in the brain were monitored using high-field magnetic resonance imaging (MRI), electrocardiography, Holter ECG monitoring, transthoracic echocardiography, and transcranial Doppler monitoring with microembolodetection, as well as several other techniques.

Ultrasound DS and transthoracic echocardiography were performed on Philips Epiq 7G (Philips Ultrasound, Bothell, WA, USA). Linear format probes with the frequency of 3–12 MHz were used to assess the state of the aortic arch branch (innominate artery; right subclavian artery; common, internal and external carotid arteries; vertebral arteries), and sector matrix probes with a frequency of 1–5 MHz were used for transcranial duplex scanning, as well as transthoracic echocardiography. The structure of the vascular wall, and the presence and severity of atherosclerotic changes signs, other intraluminal lesions, and artery geometries were evaluated; linear blood flow velocities included peak systolic velocity (Vps), end diastolic velocity (Ved), time-averaged maximum blood flow velocity, and were measured in the Doppler spectral mode (with angle correction), while resistance Pourcelot (RI) and pulsatility Gosling (PI) indices in the extra- and intra-cranial arteries, including ICA (ophthalmic and communicating segment), ACA (A1 segments), MCA (M1 and M2 segments), PCA (P1 and P2 segments), VA (V4 segments), and basilar artery, were also measured. Transthoracic echocardiography was used for heart (walls, chambers, and valves) and visible thoracic aorta segment condition estimation.

MRI was performed on Discovery 780 (GE, Boston, MA, USA) with a magnetic field induction of 3T, and a neurovascular coil was used. Brain MRI protocol included T1- and T2-weighted images, FLAIR pulse sequences (with an isotropic voxel, with a slice thickness of 1 mm), and images were weighted via magnetic susceptibility (SWAN). MRI was used to evaluate the characteristics of cerebral infarction (localization, size, prescription, and signs of impregnation) and, the presence and severity of other lesions. Multispiral CT angiography and CT perfusion of brachiocephalic arteries were performed on 128-slice Optima (GE, USA), with the intravenous contrast Ultravist (Yopromide, Bayer AG Berlin, Germany,) 370 mg of iodine/mL, and 1 mg/kg weight.

For each case, a group of experts after a thorough discussion based on the whole set of data demanded additional diagnostics which included transesophageal echocardiography, a bubble test, vascular wall MRI, special laboratory testing, etc., aiming to understand the possible causes of IS development.

Obtained data were formalized and then converted into spreadsheets. 

ECA types according to the classification proposed by J. Weibel and W.S. Fields were taken into account, implying division into tortuosity, coiling, and kinking [23]. At the same time, we did not rank the angle of kinking, and during duplex scanning, we avoided hemodynamic change assessment in the most deformed artery segment.

Statistical analysis was carried out using the SPSS Statistics 23.0 package. The null hypothesis was rejected at *p* < 0.05, and intergroup comparisons were carried out using the exact Fisher criterion, Student’s *t*-test, and the Mann–Whitney criterion, depending on the type of dependent variable. Patients with ischemic stroke history and artery elongation in our cohort did not differ from those without elongation (also with ischemic stroke) by age and concomitant diseases. Therefore, we decided not to introduce an adjustment for anamnestic and demographic parameters when comparing groups, especially since more complex statistical methods have less analysis power. In addition, we performed age-adjusted analysis using a general linear model with grouping variables as independent factors and age as a covariate; various scale variables were dependent. Overall analysis results did not change—i.e., significant differences were observed with the same variables, so we report only univariate analysis.

The study was performed with the support of State Assignment No. 056-00171-19-01, subject registration number AAAA-A19-119042590018-0, dated 29 March 2019.

## 3. Results

ECA was confirmed via instrumental methods in 200 patients (17.1%) out of 1171; among them, 98 (49%) were women aged 32 to 95 years (mean age 67 ± 11 years) and 102 (51%) were men aged 28 to 80 (60 ± 10) years. ECA was not detected in 971 (82,9%) individuals, among which 335 (34.5%) were women aged 27 to 87 years (an average of 63 ± 13 years) and 636 (65.5%) were men aged 21 to 90 years (an average of 60 ± 11 years). In the subgroup of patients with IS history and confirmed ECA, the proportion of women was higher (*p* ˂ 0.0005) compared to that in the subgroup with non-elongated ICA (49% and 31.5%, respectively).

As a result, significant differences in the incidence of unilateral or bilateral ECA in men and women were not detected (*p* = 0.665).

The incidence of different ECA types in all included individuals and in the group of patients with ECA is presented in Table 1

Combinations of different ICA elongation types were observed in 21 (18.9%) patients with bilateral ECA (not included in Table 1), of which 6 patients (5.4%) had tortuosity on one side and coiling ECA on the other side, 11 patients (9.9%) had tortuosity on one side and contralateral kinking, and 4 patients (3.6%) had kinking on one side and contralateral coiling ECA.

The incidence and localization of cerebral infarction is presented in Figure 1.

As presented in Figure 1, the localization of cerebral infarction in the comparison groups (with and without ECA) did not differ; ischemic lesions were most frequently located in the basal ganglia and less frequently located in the cerebellar hemispheres.

The prevalence of IS pathogenetic subtypes according to the TOAST criteria [24] is presented in Table 2.

The incidence of various IS pathogenetic subtypes differed slightly, and no statistically significant differences were found.

Correlation between the side of ECA (in patients with unilateral ICA) and the side of cerebral infarction in the anterior circulation have not been revealed (*p* = 0.05), while unilateral ECA was accompanied by the presence of ipsilateral infarction foci in 8.7% of patients and contralateral brain infarction in 23% of patients.

The distribution of unilateral hemispheric IS subtypes and the incidence of various types of elongation are presented in Table 3.

In patients with atherothrombotic and cryptogenic IS subtypes the proportion of tortuosity on the affected side slightly exceeded or was comparable to contralateral (“healthy”) ICA, and the percentage of unilateral kinking and coiling was less than or equal to those on the opposite side.

However, it was not possible to identify the pattern of the prevalence and localization of various ECA types. Left ICA tortuosity in patients with IS in the left hemisphere, and right ICA kinking and coiling ECA in patients with left-sided IS were registered somewhat more often. It is worth noticing that for the left-side-stroke ECA subgroup patients with lacunar stroke exclusively had tortuosity with equal frequency observed in both ipsilateral and contralateral ICA.

Other determined IS etiologies were occasionally detected in patients with ECA.

An analysis of brain MRI data did not reveal an association between the presence, type of ECA, localization of ipsilateral stroke lesions, areas of cerebral cortex regional atrophy, and the severity of white matter lesions according to Fazekas scale assessment (Figure 2a,b).

A statistically significant correlation between the presence of ECA and the BMI value was determined (BMI in with ECA, 28.2 ± 5 kg/m^2^; BMI in patients without ECA, 27.5 ± 4.9 kg/m^2^). Moreover, the incidence and severity of obesity in patients with ECA and without ECA did not significantly differ. 

Moreover, there were no statistically significant differences in the prevalence of diabetes mellitus and AH (*p* = 0.628 and *p* = 0.614, respectively); the incidence of the latter was equally high in both groups. Thus, diagnosis of AH was established in 167 patients (83.5%) with ECA, and in 795 people (81.9%) without ECA.

Doppler extracranial artery parameters (measured and calculated) are presented in Figure 3. Statistically significant decreases in the Vps in the right ICA (*p* = 0.036) and in the left ICA (*p* < 0.0005), and decreases in the end diastolic velocity (Ved) in the left ICA (*p* < 0.0005) were registered in the ECA group (Figure 3a). 

Vps decreases in A1 segments of anterior cerebral arteries in patients with ECA compared with those without ECA were also registered (on the right, *p* = 0.01; on the left, *p* = 0.06). Ved in the P2 segments of both posterior cerebral arteries (PCA) in patients with ECA were lower compared to those in individuals without ECA (on the right PCA, *p* = 0.013; on the left, *p* = 0.035). Statistical corrections for a pathogenetic subtype of stroke and (or) ischemic lesion localization did not change the pattern mentioned above.

There were differences in the flow characteristics of posterior circulation arteries (Figure 3b) with a significant decrease in Vps (*p* < 0.0003 on the right; *p* < 0.003 on the left), and Ved (*p* < 0.0005 and *p* < 0.001, respectively) in the extracranial sections of both VAs in comparison with those in patients without ECA.

Transthoracic echocardiography showed a statistically significant increase in the left atrium size in patients with ECA compared to that in individuals without ECA (68.8 ± 24 mm vs. 63.45 ± 23 mL, *p* = 0.019). Deformations of vertebral arteries along with ICA elongations were revealed. The results of various types of non-linear VA in the V1 segment and V2 segment analysis are shown in Table 4.

## 4. Discussion

The characteristics of our patient’s cohort may impose restrictions on the study results. The investigated cohort consisted of patients with an IS history. On the one hand, this was the main reason to carry out ongoing research. On the other hand, any causal relationships without confirmed IS could only be hypothetical and the evidence of the connections could be questionable. In turn, survival from IS introduces significant uncertainties associated with the probable etiopathogenetic heterogeneity of this condition, and the relativity of the possible causes, the verification of which is not always possible. Finally, critical parameters for rehabilitation—the time after IS development (up to a year) and neurological deficit severity (three points on the mRS)—could determine the specificity of our cohort, distinguishing it from a whole IS survivor population.

The results obtained in this study describe some of the constitutional as well as structural and hemodynamic features investigated in IS patients divided into subgroups by the presence or absence of ICA elongations.

Some of our results require discussion. A high prevalence of women among the patients with ECA was registered. Moreover, similar results were obtained in several studies. L. Di Pino et al., who assessed carotid arteries in 2856 persons of all age groups (from 0 to 96 years, at an average of 58 ± 22) using the DS method, showed that the proportion of women was 51.4% in patients with ECA and 41.9% in individuals without ECA; [8]. F.G. Hugo Martins et al., who conducted a clinical ultrasound examination of 19,804 persons aged over 25 years, also confirmed a predominance of women in the group with ECA, this being found in 1686 cases (63%) compared to those without ECA, at 8076 cases (47%) [25]. This fact could be partially explained by the estrogen deficiency effect. This effect leads to a decrease in the hydrophilicity of the connective tissue and collagen structures that form tunica media arteries, and the decrease in the intervertebral disc height in menopausal women. M.P. Brincat et al. came to similar conclusions when comparing these characteristics in women undergoing hormone replacement therapy for estrogen deficiency with those who had a natural course of menopause [26]. Thus, we can assume that the decrease in the cervical spine height may lead to ECA formation or an increase in the severity of existing elongation. In addition, the prevalence of arterial hypotension among women in the first decades of life is higher, whereas in older age arterial hypertension can also affect the formation of ECA [27,28].

Comparing cerebral infarction localization, no statistically significant differences were shown between groups of patients (Figure 1). The incidence of stroke subtypes according to the TOAST criteria [24] did not significantly differ between patients with ECA (both unilateral and bilateral) and the comparison subgroup.

An analysis of various IS subtypes’ prevalence revealed statistical differences between both comparison subgroups (with and without ECA), and there were no differences between patients with unilateral and bilateral arrangements of ECA (Table 2). Data obtained in similar organized cohort studies demonstrate contrasting results, which is possibly due to differences in patient inclusion/exclusion criteria and clinical and instrumental examination protocols. The number of such studies is small, and the investigation of ECA features is usually not the main goal of research. Thus, for example, among the patients included in the PARISK (Plaque at RISK) cohort clinical study selected randomly from 100 patients with IS, ICA was evaluated using CTA in 200 patients, and ECA was detected in 136 patients (68%). Tortuosity prevailed among the ECA configurations (72%), coiling was verified in 20%, and kinking was present in 8% [29]. Another study aiming to evaluate the incidence of ICA coiling and identify potential factors associated with this condition in patients with verified IS was conducted using CTA data from a multicenter randomized study, DUST (Dutch Acute Stroke Trial) [30]. Elongation was registered in 146 cases (5 cases were excluded due to poor CTA quality), and among the remaining 141 patients 108 patients had unilateral and bilateral ICA coiling (35 cases (24%) and 73 cases (76%), respectively) and tortuosity, and kinking (total) accounted for 33 patients (23%), which distinguishes these data from the study cited above [30].

Our study results (Table 2) fall between those described above. Thus, in patients with the atherothrombotic stroke subtype of ipsilateral ICA, tortuosity (right 37.7%; left 41.4%) and kinking (23.1% and 41.4%, respectively) were more common. In patients with the cardioembolic stroke subtype, the incidence of ICA coiling was higher compared to that in patients with atherothrombotic stroke, and tortuosity in these patients was registered less often. In patients with cryptogenic IS, the incidence of various ECA types was almost identical.

Both of the studies cited above and our data could not emphasize the predominance of any particular ECA type among stroke survivors. Discrepancies in different types of ECA incidence can be explained by the differences between applied instrumental imaging methods. In our study, in contrast to studies based exclusively on CT data [29,30] we used CTA, MR angiography, and (mainly) DS, the least "sensitive" method in ECA diagnosis, which is due to the common localization of arterial deformities well outside of the confident scanning field, and due to the "screening" by bone structures.

The abovementioned demonstrates that using several imaging diagnostic methods with different resolutions in the diagnosis of ECA without uniform estimation criteria causes significant discrepancies in the final data and imposes limitations on the conclusions.

When determining blood flow characteristics using DS, we observed a decrease in linear flow velocities in the precerebral and intracranial arteries in patients with ECA (Figure 3a,b). These results agree with those of our previously published study investigating brain blood supply based on multislice CT perfusion data [31]. In this study, slight decreases in blood flow parameters (CBF, CBV, and Tmax) were recorded in ACA circulation, the border zone of the ACA and MCA, and lenticulostriate arteries in patients with ECA in comparison with those of patients without ECA.

Regardless of the decrease in the extracranial arteries’ Vps and cerebral perfusion values shown via different imaging methods, this cannot be considered direct proof of the ECA’s negative effect on cerebral blood flow. There are some serious reasons for this statement. CT perfusion map construction is contingent on the time delay between the contrast bolus arriving at the proximal arterial circulation vessel and brain parenchyma, regardless of its causes [32,33]. In patients with ECA the a decrease in cerebral perfusion can be explained by both (Vps and Ved) lower blood flow velocities, the prolongation of the contrast agent delay, and a possible combination of these factors. We suppose that primary elongation with an increased absolute length of the artery is naturally associated with a more pronounced prolongation of the contrast delay compared to secondary type, when the distance covered by the blood does not change, and elongation is relative.

The decrease in the linear flow velocities in extracranial arteries and intracranial arteries both in anterior and posterior circulations has no independent significance. Therefore, it is not very pronounced, and it does not directly correlate with brain perfusion. 

A separate analysis of the data in patients with an atherothrombotic pathogenic variant of IS (according to TOAST) showed insignificant differences in flow velocity parameters in the anterior circulation arteries. We have not received evidence of a greater severity of small vessel disease (SVD) in the group with ECA. On the contrary, the incidence of Fazekas 3 was slightly higher in patients without ECA (Figure 2a,b). The prevalence of AH in both groups was the same, and the blood flow velocities in ICA and intracranial arteries were lower in the ECA group. It is believed that the infiltration of the brain blood barrier (BBB) is one of the main causes of SVD [34].

Hypothetically, it can be assumed that ECA can serve as a certain “protection” component of the BBB system from pulse wave traumatization following blood pressure increases, preventing an increase in the permeability of the barrier and a pathological cascade leading to astrogliosis and decelerating the development of small vessel disease and related clinical manifestations, such as cognitive impairment [35]. Our hypothesis, however, needs experimental evidence.

At the same time, in the posterior circulation arteries (VA and PCA) in the ECA group, blood flow velocities were significantly lower than those in the comparison group. Thus, carotid artery stenotic atherosclerosis (a stenosis of ipsilateral arteries of more than 50% in diameter) has a greater effect on cerebral blood flow than ECA does, and the hemodynamic effect of the latter becomes even less obvious (Figure 3a,b). It should be also considered that a possible decrease in blood flow on the side of infarction, collateralization with high degrees of ICA stenosis and occlusions, and the effect of intracranial artery stenosis were ignored in these subgroup analysis data. 

An investigation of the relationship between the localization and types of ECA and the pathogenic subtypes of IS (Table 3) showed a slightly higher proportion of ipsilateral ICA tortuosity in patients with the atherothrombotic IS subtype compared to that on the contralateral side. At the same time, the proportion of kinking was smaller, which imposes certain restrictions on the conclusions about ECA as a potentially embologenic condition.

For patients with the verified cardioembolic subtype of IS (according to TOAST), it was not possible to trace any consistency in the frequency of certain ECA types; however, kinking and coiling deformities of the right (ipsilateral) ICA occurred somewhat more often than did tortuosity on the same side and the opposite side (right ICA: tortuosity, 12.5%; coiling, 18.7%; kinking, 29.6%; left ICA: 24%; 8.3%; 8.7%; respectively). At the same time, the proportion of cardioembolic IS in the ECA subgroup (Table 2) was greater than that in the comparison group (18.5% and 11.4%, respectively). Since no data demonstrate the predominant ways emboli move in thrombogenic pathologies of the heart and aorta, it is difficult to explain these findings in our cohort.

Nevertheless, the "protective" influence of kinking (as well as atherosclerotic stenosis of high grades) cannot be excluded. The well-known fact of an increase in peripheral resistance proximal to "significant" obstacles for blood flow (ECA, stenosis of more than 60% in diameter) with the following redirection of emboli to other branches with lower flow resistance can stand in the attempt to explain the above hypothesis. 

It is noteworthy that there were no differences between certain types of ECA incidence (Table 3) in patients with cryptogenic IS (45.5% among persons with ECA and 46.3% without ECA, Table 2), in contrast to the patients described above with atherothrombotic and cardioembolic IS variants. Coiling deformations and tortuosties of ipsilateral ICA were some more common with a right–sided location of infarction (61.5% and 51.5%, respectively), and in patients with left-sided stroke, left ICA tortuosity accounted for 45%, coiling accounted for 43.8%, and kinking accounted for 33.3%.

Consequently, in the group of patients with cryptogenic IS, there was the largest amount of kinking (among similar configurations of all ECAs). There is an assumption, which is partially confirmed by the results of pathomorphological studies, that there is a greater risk of dissections in patients with ICA kinking due to the restructuring of smooth muscle elements, the elastic framework of failed vascular walls and the micro–ruptures of its intima [36]. In our study, we were unable to confirm this; ICA dissection was diagnosed only once in patients with ECA while in 971 patients from the group without ECA 5 ICA dissections were verified via visualization methods (i.e., in both groups, the frequency of ICA dissection detection was similar and amounted to about 1:200).

One of the study’s objectives, apart from the correlation between the presence of ECA and IS, was identifying potential factors that could lead to ECA formation. Scientists who adhere to the secondary ECA origin hypothesis consider obesity (an increase in BMI values) as one of the most significant risk factors which leads to the relative secondary elongation of large vessels [37]; however, such assumptions were not confirmed in our study.

In addition to the issue of the possible effect of ECA on cerebral blood flow, the question of its origin remains debatable. If ECA is observed in patients with symptoms of genetically determined connective tissue disorder development, they are considered to be congenital. However, the number of such syndromes being verified is far fewer than the incidences of ECA. For example, the incidence of detecting the “vascular” subtype of Ehlers–Danlos syndrome is 1 case per 90,000 people (the incidence of all subtypes of this syndrome is 1:20,000 [38], while the incidence of Marfan syndrome is 17:100,000 [39]. The incidence of Loeys–Dietz syndrome has not been described yet, due to the small amount of data [40]. As already mentioned, the incidence of ECA can reach up to 58% in both children and adults. Changes in the configuration of carotid arteries during active skeletal growth in children are also described [8]. Therefore, the formation of ECA cannot be explained solely by interruptions of connective tissue development.

The most convincing of the few studies investigating the relationship between ECA formation and arterial hypertension (AH) is the one performed by R. Pancera et al. [6]. When analyzing arterial kinking frequency between normotensive patients and persons with hypertension, no significant difference was found for bends of 90° and 60°–30°, and a statistically significant difference (*p* ˂ 0.02) was registered for prominent rough bends (angle less than 30°). At the same time, the authors stated the absence of a correlation between ECA and neurological symptom severity; however, they showed a link (*p* ˂ 0.01) between ECA presence and TIA. At the same time, the incidence of ECA in IS patients does not differ from that in patients without IS [4]. Similar results related to coiling ECA (so-called coiling) were obtained by J.L.M. van Rooij et al. when analyzing data from the multicenter randomized DUST study already mentioned above [30]. The authors compared two groups, one with patients that had ECA (coiling) and IS anamnesis, and the second being similar in key parameters, but without ECA, which was selected via the case–control method. There was a statistically significant predominance of patients with hypertension in the ECA group (*p* = 0.033). Uni- and multi-variate analysis showed a correlation between ECA and hypertension (*p* = 0.034, this link remained significant with adjustments for gender, age, and the presence of ICA atherosclerosis (one by one or together)) [30]. The most rational explanations for the difference between the results of our study and the studies mentioned above may be the following. First, we considered only cases with an officially documented diagnosis of hypertension in IS survivors; second, our study was multimodal and all cases with ECA were verified predominantly with the DS BCA, which may lead to insufficient ECA patient coverage. Both factors may have affected our results.

We presented a statistically significant increase in the left atrium size as measured via echocardiography in patients with ECA. Such changes, on the one hand, can correspond to hypertension consequences, although there was no difference in the incidence of hypertension between comparison groups. On the other hand, it is impossible to ignore the probability of congenital feature presences associated with connective tissue structure, which can lead to fibrous ring dilatation and mitral valve insufficiency formation with a subsequent increase in the atrium volume or a combination of these factors [41].

We described an increase in the size of the left atrium in patients of both groups, but in the ECA group, the sizes were statistically significantly bigger than those in patients without ECA. The atrium enlargement of the patients included in our study, i.e., stroke survivors, is a common find, and heart pathology can be the cause of ischemic stroke as well as being observed along the way, due to the presence of common risk factors for cardiovascular pathology. Between the ECA group and in patients without ECA, the incidences of the cardioembolic IS subtype were comparable. There were no statistically significant differences in the incidence of AH, and the average age of the patients was comparable. Thus, there were no statistically significant intergroup differences that could have artificially influenced the distribution of patients according to the LA size criterion.

We assume that structural changes leading to an increase in LA volume in patients with ECA, in addition to diseases of the cardiovascular system, may be caused by genetically determined morphological features of the elastic framework of the atrium, which can lead to the stretching of the annulus fibrosus, infractions of the mitral valve, etc.

Deformations of the vertebral arteries are also worth mentioning. The lack of established methodological approaches for their classification makes it difficult to systematize such conditions. Corresponding to the ultrasound data, the incidence of the VA deformations was comparable both in patients with ECA and in the group of patients without ICA deformities. The latter may be evidence of differences in mechanisms that lead to VA and ICA deformities (primary and secondary elongation) or their combination in some cases, which was acknowledged by several authors [42]. On the other hand, VA elongation can also occur due to other conditions, such as degenerative changes in cervical and thoracic sections of the spine, which lead to a relative increase in artery length [43].

We did not evaluate the acute IS treatment methods used in the patients included in the study. The main goal was to find the relationship between the presence of ECA and the subtype and localization of IS. We did not succeed. However, there is a consolidated and unconditional point of view, according to which the presence of disturbances in the geometry of the precerebral arteries significantly complicates endovascular interventions. Rough kinking limits the choice of equipment, increases the manipulation time and thus can affect the outcome of the disease as described in Pierre K. et al. [44].

There is another important circumstance that is ignored in most studies. When investigating ECA, researchers took into account and analyzed only arteries; tortuosity, coiling, and kinking (with a gradation of angle value). However, it is obvious that such arterial changes are not limited to the ones mentioned above. The number of arterial deformity types may be significantly higher, and logical relationships may be various for distinct variants. Thus, an analysis of the nature, prevalence, and severity of ECA, as well as determining factors influencing its formation, regardless of their hemodynamic and (or) pathogenetic significance, is an essential task for the future.

## 5. Conclusions

In patients with IS anamnesis, ECA was significantly more common in women than men (*p* < 0.0005);Patients with and without ECA did not statistically significantly differ in terms of stroke localization (anterior or posterior circulation) and IS subtype frequency.There was no obvious link between ECA and well-known cardiovascular risk factors (BMI, DM, and AH);An association between ECA localization, the side of ischemic stroke, well, and stroke pathogenetic variant was not registered;Linear blood flow velocities in some extra- and intra-cranial arteries were lower in patients with ECA compared to those in the group without ECA;The elongation of the carotid arteries in the examined patients was associated with vertebral artery deformations, which manifested in the form of a decrease in linear blood flow velocities in the extra- and intra-cranial arteries’ peak systolic flow velocity in the ICA and VA on both sides, end diastolic velocity in the left ICA, the A1 segments of both ACAs, and P2 segments of both PCAs.

## Figures and Tables

**Figure 1 biomedicines-11-02751-f001:**
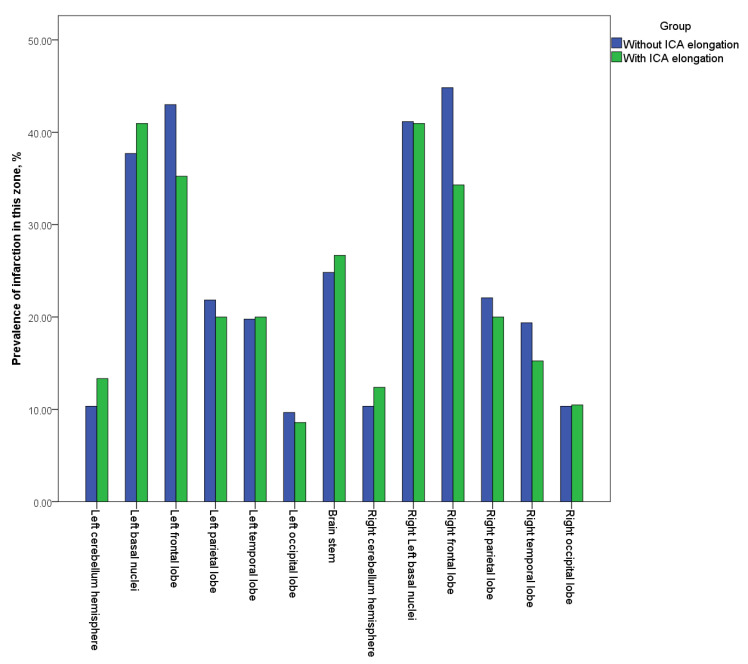
Localization of ischemic lesions.

**Figure 2 biomedicines-11-02751-f002:**
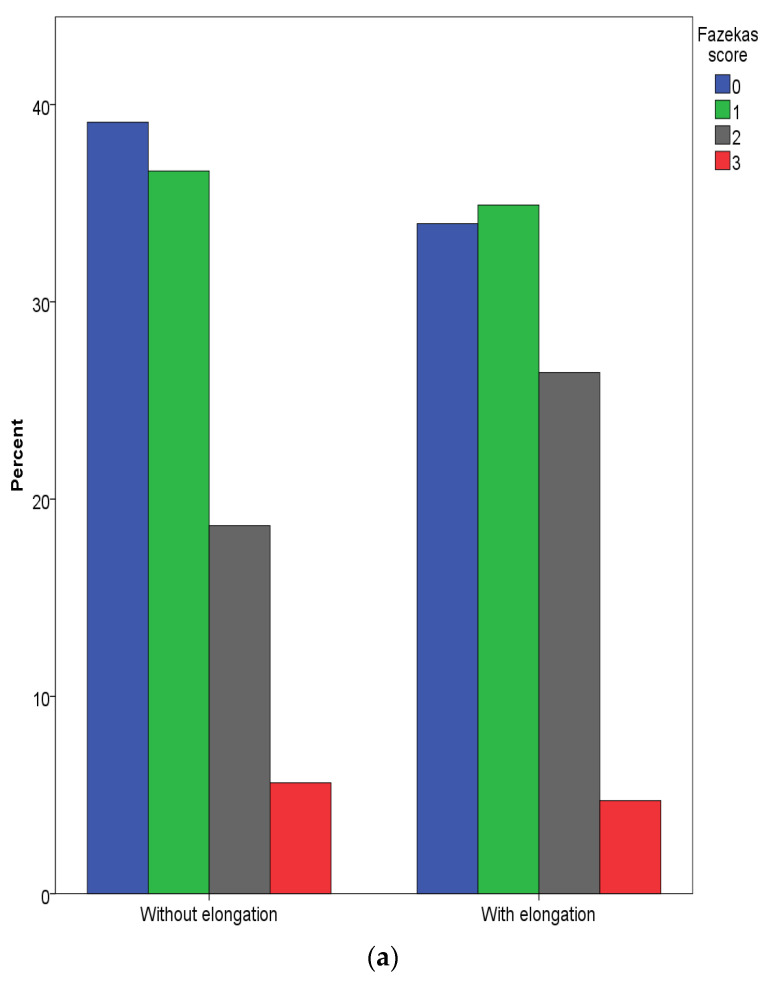
(**a**) The severity of periventricular white matter lesions assessed using the FAZEKAS scale. (**b**) The severity of deep white matter lesions assessed using the FAZEKAS scale.

**Figure 3 biomedicines-11-02751-f003:**
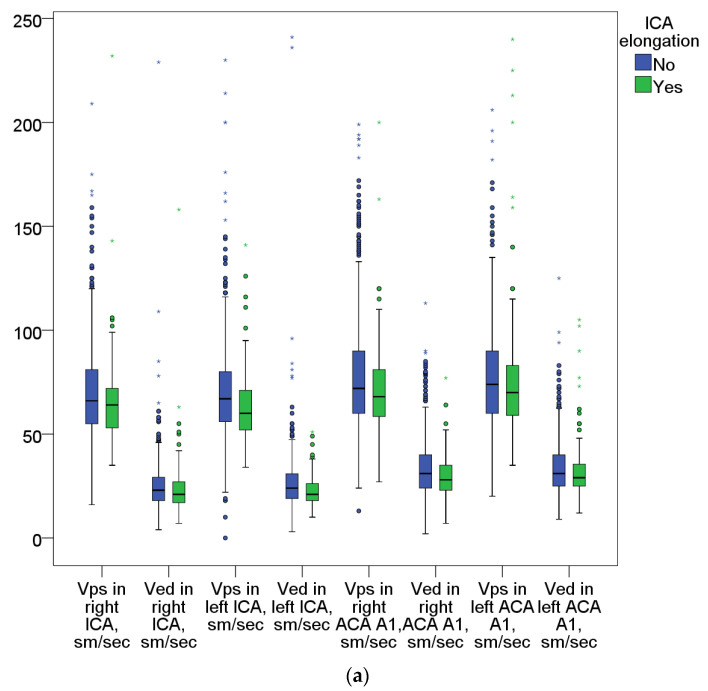
(**a**) Peak systolic and end diastolic blood flow velocities in the anterior circulation arteries. (**b**) Peak systolic and end diastolic blood flow velocities in the posterior circulation arteries. Note: circles: outliers of more than a 1.5 IQR from the upper (or lower) quartile; asterisk: outliers of more than a 2 IQR from the upper (or lower) quartile.

**Table 1 biomedicines-11-02751-t001:** Frequency of different ECA types.

ECA Side	Type of ECA					
	Tortuosity		Coiling or Other Complex Shape		Kinking ≤ 90°	
	1	2	1	2	1	2
Right ICA	20 (45.5%)	10.0%	9 (20.5%)	4.5%	15 (34.1%)	7.5%
Left ICA	19 (42.2%)	9.5%	12 (26.7%)	6.0%	14 (31.1%)	7.0%
Both ICA	48 (43.2%)	24.0%	7 (6.3%)	3.5%	35 (31.5%)	17.5%

Note: 1: frequency, absolute number of cases (percentage of patients with this side of deformity, %); 2: frequency among all patients with ECA, %.

**Table 2 biomedicines-11-02751-t002:** Prevalence of subtypes of ischemic stroke (TOAST) in patients of comparison groups.

Pathogenetic Subtypes of IS (by TOAST [24])	ECA				
	Yes			No (807 People)	
	One Side (89 People)	Both Sides (111 People)	Total (200 People)		
	Absolute Quantity (Percentage of Total Quantity)				
Atherothrombotic (n = 436)	25 (28.1%)	38 (34.2%)	63 (31.5%)	373 (38.4%)	
Cardioembolic (n = 148)	21 (23.6%)	16 (14.4%)	37 (18.5%)	111 (11.4%)	
Lacunar (n = 37)	2 (2.2%)	6 (5.4%)	8 (4.0%)	29 (3%)	
Other determined etiology (n = 9)	Hemodynamic	1 (1.1%)	0	1 (0.5%)	6 (0.6%)
	Venous	0	0	0	1 (0.1%)
	Other causes	0	0	0	1 (0.1%)
	Total	40 (44.9%)	51 (45.9%)	91 (45.5%)	450 (46.3%)
Stroke of unknown etiology/Cryptogenic stroke (n = 541)	25 (44.6%)	31 (45.6%)	70 (43.8%)	373 (46.2%)	

**Table 3 biomedicines-11-02751-t003:** Incidence of ECA in patients with different subtypes of IS (TOAST [24]).

Pathogenetic Subtypes of IS (TOAST)	Stroke Localization											
	Left (n = 97; 48.5%)						Right (n = 81; 40.5%)					
	RICA Elongation Type (Contralateral)			lICA Elongation Type (Ipsilateral)			RICA Elongation Type (Ipsilateral)			lICA Elongation Type (Contralateral)		
	Tortuosity	Coiling (Other Configurations)	Kinking (≤90°)	Tortuosity	Coiling (Other Configurations)	Kinking (≤90°)	Tortuosity	Coiling (Other Configurations)	Kinking (≤90°)	Tortuosity	Coiling (Other Configurations)	Kinking (≤90°)
Atherothrombotic	11 (29.7%)	4 (33.3%)	12 (37.5%)	24 (41.4%)	2 (15.4%)	12 (41.4%)	15 (37.5%)	3 (18.7%)	6 (23.1%)	12 (36.4%)	2 (16.7%)	10 (43.5%)
Other determined etiology	0 (0.0%)	0 (0.0%)	0 (0.0%)	0 (0.0%)	0 (0.0%)	0 (0.0%)	0 (0.0%)	1 (6.3%)	0 (0.0%)	1 (3.1%)	0 (0.0%)	0 (0.0%)
Cardioembolic	9 (24.3%)	4 (33.3%)	4 (12.5%)	11 (19.0%)	3 (23.1%)	2 (6.9%)	5 (12.5%)	3 (18.7%)	7 (26.9%)	8 (24.2%)	1 (8.3%)	2 (8.7%)
Stroke of undetermined etiology/Cryptogenic stroke	13 (35.1%)	4 (33.3%)	16 (50%)	18 (31.0%)	8 (61.5%)	15 (51.7%)	18 (45.0%)	7 (43.8%)	12 (33.3%)	11 (31.8%)	8 (66.7%)	11 (47.8%)
Lacunar	4 (10.8%)	0 (0.0%)	0 (0.0%)	5 (8.6%)	0 (0.0%)	0 (0.0%)	2 (5.0%)	2 (12.5%)	1 (3.0%)	1 (4.5%)	1 (8.3%)	0 (0.0%)

**Table 4 biomedicines-11-02751-t004:** Frequency of vertebral artery elongation types (segments V1 and V2) in patients with and without ECA.

DCA	Right VA			Left VA		
	V1 Segment Tortuosity					
	Severe	Moderate	No	Severe	Moderate	No
No	1 (0.8%)	40 (32.8%)	81 (66.4%)	15 (12.2%)	54 (43.9%)	54 (43.9%)
Yes	4 (6.2%)	24 (36.9%)	37 (56.9%)	8 (13.6%)	28 (47.5%)	23 (39.0%)
	V2 Segment tortuosity					
No	5 (4.0%)	27 (21.8%)	92 (74.2%)	3 (2.5%)	31 (25.6%)	87 (71.9%)
Yes	2 (3.1%)	16 (24.6%)	47 (72.3%)	2 (3.3%)	15 (24.6%)	44 (72.1%)

## Data Availability

The data presented in this study are available on request from the corresponding author.

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
