# Peer review of "Carotid Dolichoarteriopathy (Elongation) of the Carotid Arteries in Patients with Ischemic Stroke Anamnesis"

_biomedicines, 2023, doi:10.3390/biomedicines11102751_

Round 1

Reviewer 1 Report

Reviewing the manuscript entitled, “Carotid dolichoarteriopathy (elongation or deformation) of the carotid arteries people who have suffered an ischemic stroke”, this is a retrospective cohort study focusing on relevance between dolichoarteriopathy of the carotid artery and stroke. Although this is a clinical study, The results are mostly negative data, and statistical analysis methods are not appropriate. This is extremely difficult to read due to poor text structure.

In the introduction section, the first of all the authors have to do that is description of definition of carotid dolichoarteriopathy. In addition, the authors need to describe the epidemiology and hemodynamics.

Generally, it is thought that dolichoarteriopathies of the carotid artery may reduce the blood supply to the brain through decreases in blood pressure, which often do not lead to cerebral ischemia due to compensation of the self-regulatory mechanism in the cerebral blood supply. Nonetheless, the authors should describe the circumstances that led to the study of the link between dolichoarteriopathies of the carotid artery and stroke.

ESA and stroke are strongly influenced by lifestyle, age, cardiovascular disease, and metabolic disease. Thus, such confounding factors affect statistical analysis. If so, I think the statistical analysis method used by the author is suboptimal.

In the results section, the authors mentioned that “Among those who underwent IS with ECA, proportion of women is higher (p˂0,0005) than in the subgroup with non-elongated ICA (49% and 31,5%, respectively).”. Which table or graph is this? Data that show a significant difference are important data. The authors should be shown in graphs or tables.

Left atrial diameter shows left atrial enlargement in both groups. Is it possible that both groups had subjects with cardiovascular disease?

Is there no association with conclusion 6 with age?

The manuscript is extremely difficult to read because the content of the text does not match the graphs and tables.

Author Response

Answer to reviewer 1

  1. In the introduction section, the first of all the authors have to do that is description of definition of carotid dolichoarteriopathy. In addition, the authors need to describe the epidemiology and hemodynamics.

 As the title of the article suggests, "Dolichoarteriopathy" is a synonym for elongation. There are no specific definitions. Due to the lack of clear diagnostic criteria, the use of various imaging modalities, as well as the different occurrence in people of different ages, the prevalence of dolichoaretriopathy will depend on the combination of these components. The frequency of occurrence of dolichoarteripathy in different authors is given in the range of 10-58% [Schenk P, Temmel A, Trattnig S, Kainberger F (1996) Current aspects in diagnosis and therapy of carotid artery kinking.HNO 44:178–185, Saba L, Argiolas GM, Sumer S, et al. Association between internal carotid artery dissection and arterial tortuosity. Neuroradiology 2015;57:149–53; Beigelman R, Izaguirre AM, Robles M, Grana DR, Ambrosio G, Milei J. Are kinking and coiling of carotid artery congenital or acquired?. Angiology. 2010;61(1):107-112. doi:10.1177/0003319709336417; Khasiyev F, Gutierrez J. Cervical Carotid Artery Dolichoectasia as a Marker of Increased Vascular Risk. J Neuroimaging. 2021 Mar;31(2):251-260. doi: 10.1111/jon.12815. Epub 2020 Nov 27. PMID: 33244825].

Assessment of hemodynamics in carotid dolichoarteriopathy is usually reduced to measurements of peak systolic blood flow velocity in an indirect line segment, by means of ultrasound duplex scanning, which is characterized by a large operator dependence. As an example, I cite the values of the peak systolic blood flow rate, which is regarded as a threshold for making a decision on the clinical significance of dolichoarteriopathy: 150 -250 сm/s [Usachev DYu, Lukshin VA, Sosnin AD, et al.  Surgical management of patients with pathological deformation of carotid arteries. J Probl of neurosurgery named after N.N. Burdenko. 2014;5:3-15. (In Russ.) Усачев Д.Ю., Лукшин В.А., Соснин А.Д. и др. Хирургическое лечение больных с патологическими деформациями сонных артерий. Вопросы нейрохирургии имени Н.Н. Бурденко. 2014;5:3-15. ISSN 0042-8817, Wang L, Zhao F, Wang D, et al. Pressure Drop in Tortuosity/Kinking of the Internal Carotid Artery: Simulation and Clinical Investigation. Biomed Res Int. 2016;2016:2428970. doi:10.1155/2016/2428970.; Sarkari NB, Holmes JM, Bickerstaff ER. Neurological manifestations associated with internal carotid loops and kinks in children. J Neurol Neurosurg Psychiatry. 1970;33(2):194-200. doi: 10.1136/jnnp.33.2.194; La Barbera G, La Marca G, Martino A, et al. Kinking, coiling, and tortuosity of extracranial internal carotid artery: is it the effect of a metaplasia? Surg Radiol Anat. 2006;28(6):573-80. doi: 10.1007/s00276-006-0149-1; Illuminati G, Ricco JB, Caliò FG, et al. Results in a consecutive series of 83 surgical corrections of symptomatic stenotic kinking of the internal carotid artery. Surgery. 2008;143(1):134-9. doi: 10.1016/j.surg.2007.07.029.],  Therefore, in this work we avoid hemodynamic characteristics. In addition, given the presence of an autoregulation system of cerebral circulation, local hemodynamic disorders in the precerebral arteries can be ignored.

We partially added this information in manuscript.

  1. Generally, it is thought that dolichoarteriopathies of the carotid artery may reduce the blood supply to the brain through decreases in blood pressure, which often do not lead to cerebral ischemia due to compensation of the self-regulatory mechanism in the cerebral blood supply. Nonetheless, the authors should describe the circumstances that led to the study of the link between dolichoarteriopathies of the carotid artery and stroke.

In addition to the hypothesis of a decrease in the level of blood supply to the brain in patients with dolichoarteriopathy, there is also a theory of embolism based on the assumption of an incomplete elastic framework of the wall of the non-linear segment and associated micro-injuries of intima with the formation of subintimal hematomas [Usachev DYu, Lukshin VA, Sosnin AD, et al.  Surgical management of patients with pathological deformation of carotid arteries. J Probl of neurosurgery named after N.N. Burdenko. 2014;5:3-15. (In Russ.) Усачев Д.Ю., Лукшин В.А., Соснин А.Д. и др. Хирургическое лечение больных с патологическими деформациями сонных артерий. Вопросы нейрохирургии имени Н.Н. Бурденко. 2014;5:3-15. ISSN 0042-8817].  In this work, we did not set out to prove or disprove these theories. Our task was to compare two groups of patients who had an ischemic stroke with and without dolichoarteriopathy.

  1. ESA and stroke are strongly influenced by lifestyle, age, cardiovascular disease, and metabolic disease. Thus, such confounding factors affect statistical analysis. If so, I think the statistical analysis method used by the author is suboptimal.

In our cohort, patients with ischemic stroke history and artery elongation did not differ from those without elongation (also with ischemic stroke) by age and concomitant diseases. Therefore, we decided not to introduce an adjustment for anamnestic and demographic parameters when comparing groups, especially since more complex statistical methods will have less analysis power. We tried to compare the groups age-adjusted, but the analysis results did not change.

  1. In the results section, the authors mentioned that “Among those who underwent IS with ECA, proportion of women is higher (p˂0,0005) than in the subgroup with non-elongated ICA (49% and 31,5%, respectively).”. Which table or graph is this? Data that show a significant difference are important data. The authors should be shown in graphs or tables.

We decided not to complicate the perception of the text of the article with unnecessary tables and therefore pointed out the revealed difference in the number of women in the comparison groups in text format since it's just a couple of numbers. We used the tables exclusively to demonstrate the similarity or difference of the set of parameters being compared, where we considered it necessary.

  1. Left atrial diameter shows left atrial enlargement in both groups. Is it possible that both groups had subjects with cardiovascular disease?

Patients with cardiovascular diseases were present in both groups and did not differ significantly in its frequency. All patients included in the study suffered an ischemic stroke. The size of the left atrium in patients with dolichoarteriopathy turned out to be larger, but this is not enough for decisive conclusions. To confirm this fact, studies on other cohorts of patients and a different study design are needed.

  1. Is there no association with conclusion 6 with age?

Tortuosity of the vertebral arteries and a decrease in blood flow rates in the precerebral arteries and brain arteries often correlate with age, however, in this study, the patients of the comparison groups did not differ significantly in this characteristic

Our group is grateful to reviewer 1 for a detailed analysis and valuable comments!

Reviewer 2 Report

This is a very interesting topic and has clinical merit; however, the manuscript requires a re-write/editing

This manuscript requires a re-write/editing.  There are some incomplete and run-on sentences that need attention.  The authors need to focus on sentence structure, grammar, verb tense, punctuation and there are missing words, such as "the", "in", "and", "an", etc. in many sentences and when these words are added in the sentence, the sentence will have better clarification.  I strongly urge the authors to consider a re-write/editing of the manuscript for this is an interesting topic that has clinical merit.  I would recommend that the authors have an independent individual review the revision before submitting the revised manuscript to the journal for consideration for publication.  I am willing to review the revised manuscript once it has been submitted to the journal.

Author Response

Answer to reviewer 2

We apologize for the English language translation quality. We arranged an external review and this significantly improved the translation quality and data perception (we hope). Also our group is very grateful to reviewer 2 for a valuable comments!

Round 2

Reviewer 1 Report

Reviewing the manuscript entitled, “Carotid dolichoarteriopathy (elongation or deformation) of the carotid arteries people who have suffered an ischemic stroke”, this is a retrospective cohort study focusing on relevance between dolichoarteriopathy of the carotid artery and stroke. You partially responded to my concerns, but you mentioned that “In our cohort, patients with ischemic stroke history and artery elongation did not differ from those without elongation (also with ischemic stroke) by age and concomitant diseases. Therefore, we decided not to introduce an adjustment for anamnestic and demographic parameters when comparing groups, especially since more complex statistical methods will have less analysis power.” from line 154 to line 159 in the Material and Methods section. The results of statistical analysis performed without considering the influence of confounding factors are just statistical analysis results, and medical interpretation based on them is difficult. The contents of the discussion section are statistical analysis results obtained by univariate analysis such as the student's T-test, and do not reflect medical significance.

 Left atrial diameter shows left atrial enlargement in both groups. Is it possible that both groups had subjects with cardiovascular disease? The authors should discuss firmly in the discussion section about left atrial enlargement disease and its consequences. Both groups show outliers.

Author Response

  • Reviewing the manuscript entitled, “Carotid dolichoarteriopathy (elongation or deformation) of the carotid arteries people who have suffered an ischemic stroke”, this is a retrospective cohort study focusing on relevance between dolichoarteriopathy of the carotid artery and stroke. You partially responded to my concerns, but you mentioned that “In our cohort, patients with ischemic stroke history and artery elongation did not differ from those without elongation (also with ischemic stroke) by age and concomitant diseases. Therefore, we decided not to introduce an adjustment for anamnestic and demographic parameters when comparing groups, especially since more complex statistical methods will have less analysis power.” from line 154 to line 159 in the Material and Methods section. The results of statistical analysis performed without considering the influence of confounding factors are just statistical analysis results, and medical interpretation based on them is difficult. The contents of the discussion section are statistical analysis results obtained by univariate analysis such as the student's T-test, and do not reflect medical significance.

Thank you very much for your review!

As we mentioned in previous answer to reviewer – “We tried to compare the groups age-adjusted, but the analysis results did not change.” So in order to clarify our reasons to report only univariate analysis, we added this text in material and methods section:
Besides we performed age-adjusted analysis using general linear model with grouping variable as independent factor and age as covariate; various scale variables were depend-ent. Overall analysis results did not change – i.e. significant differences were observed with the same variables, so we report only univariate analysis.

  • Left atrial diameter shows left atrial enlargement in both groups. Is it possible that both groups had subjects with cardiovascular disease? The authors should discuss firmly in the discussion section about left atrial enlargement disease and its consequences. Both groups show outliers.

To clarify our reasons, we somewhat expanded our discussion devoted to this finding:

We described an increase in the size of the left atrium in patients of both groups, but in the ECA group, the sizes were statistically significantly larger than those without ECA. Atrium enlargement in patients included in our study, i.e. stroke survivors are a frequent feature, heart pathology can be the cause of ischemic stroke as well as be observed along the way, due to the presence of common risk factors for cardiovascular pathology. In the ECA group and without ECA, the incidence of cardioembolic IS subtype was comparable. There were no statistically significant differences in the incidence of AH, the average age of the patients was comparable. So, there were no statistically significant intergroup differences that could artificially influence the distribution of patients according to the LA size criterion.

We suggest that structural changes leading to an increase in LA volume in patients with ECA, in addition to diseases of the cardiovascular system, may be due to genetically determined morphological features of the elastic framework of the atrium, which can lead to stretching of the annulus fibrosus, violations of the mitral valve etc.

Reviewer 2 Report

The authors have made some substantial improvements in this revision of their original manuscript; however, further improvements are required to be considered for publication.  There are still run-on sentences and some incomplete sentences that need attention.  The authors should still focus on sentence structure, grammar, punctuation, verb tense and to provide missing words such as "the", "an", "and", "with", etc. in some sentences.  I strongly feel that this topic has significant merit; and therefore, I urge the authors to revise the manuscript again.  I would recommend again that the authors have an independent individual review the next revision before submitting that revision to the journal for consideration for publication.  I look forward in reviewing the next revision of this manuscript.  

Attention should be towards sentence structure, such as grammar, sentence formulation, verb tense and punctuation.

Author Response

We tried to correct the manuscript English language of once more! We hope that we managed to reach an acceptable level!

In any case - thank you very much - such a look from the outside is always very useful!

Round 3

Reviewer 1 Report

This is an acceptable quality. Congrats!

Author Response

Thank you мукн much for your valuable comments!

Reviewer 2 Report

The revised manuscript has improved; however, the revised manuscript still requires attention.  There are some run-on and incomplete sentences.  Focus should also be directed to sentence structure, grammar, punctuation and verb tense.  I urge the authors to strongly consider performing another revision of the manuscript for this topic has both clinical interest and merit.  I would also recommend that the authors have an independent individual review the new revision before re-submitting the revision to the journal for consideration for publication.  I would be willing to review the new revision once it has been submitted to the journal.

The manuscript still requires attention.  There are run-on and incomplete sentences.

Author Response

Thank you for your attention to our work!
We have tried to improve the quality of the English language in our work once more, we hope we've succeeded